# Inhalation of Ultrafine Carbon Black-Induced Mitochondrial Dysfunction in Mouse Heart Through Changes in Acetylation

**DOI:** 10.3390/cells14211728

**Published:** 2025-11-04

**Authors:** Rahatul Islam, Jackson E. Stewart, William E. Mullen, Dena Lin, Salik Hussain, Dharendra Thapa

**Affiliations:** 1Division of Exercise Physiology, Department of Human Performance, School of Medicine, West Virginia University, 1 Medical Center Drive, P.O. Box 9227, Morgantown, WV 26506, USA; ri00005@mix.wvu.edu (R.I.); jes00050@mix.wvu.edu (J.E.S.); wem00001@mix.wvu.edu (W.E.M.); dl00047@mix.wvu.edu (D.L.); 2Mitochondria, Metabolism & Bioenergetics Working Group, School of Medicine, West Virginia University, Morgantown, WV 26506, USA; 3Department of Physiology, Pharmacology & Toxicology, School of Medicine, West Virginia University, Morgantown, WV 26506, USA; salik.hussain@hsc.wvu.edu; 4Center for Inhalation Toxicology (iTOX), School of Medicine, West Virginia University, Morgantown, WV 26506, USA

**Keywords:** metabolism, mitochondria, reactive oxygen species, lysine acetylation, carbon black

## Abstract

Air pollution, particularly from fine and ultrafine particulate matter (PM), has been increasingly associated with cardiovascular diseases. Ultrafine carbon, a component of ultrafine PM widely used in industrial settings, is both an environmental and occupational hazard. But the cardiac toxicity of repeated inhalation exposure to ultrafine carbon black (CB) remains unclear. In this study, we investigated how repeated inhalation of CB affects cardiac mitochondrial function, focusing on metabolic pathways and regulatory mechanisms involved in energy production. Male C57BL/6J mice were exposed to either filtered air or CB aerosols (10 mg/m^3^) for four consecutive days. Cardiac tissues were collected and analyzed to assess changes in metabolic enzyme activity, protein expression, and mitochondrial function using Western blotting, enzymatic assays, and immunoprecipitation. Despite there being few changes in overall protein expression levels, we observed significant impairments in fatty acid oxidation, increased glucose oxidation, and disrupted electron transport chain (ETC) supercomplex assembly, particularly in Complexes III and IV. These changes were accompanied by increased hyperacetylation of mitochondrial proteins and elevated levels of GCN5L1, a mitochondrial acetyltransferase. We also found increased lipid peroxidation and hyperacetylation of antioxidant enzyme SOD2 at the K-122 site, which reflects reduced enzymatic activity contributing to oxidative stress. Our findings suggest that repeated CB inhalation leads to mitochondrial dysfunction in the heart by dysregulating substrate utilization, impairing ETC activities, and weakening antioxidant defenses primarily through lysine acetylation. These findings reveal a potential role of key post-translational mechanisms in environmental particulate exposure to mitochondrial impairment and provide a potential therapeutic target for CB-induced cardiotoxicity.

## 1. Introduction

Air pollution remains a leading global environmental health risk, contributing to an estimated 4.2 million premature deaths annually, a significant proportion of which is attributed to cardiovascular diseases (CVD), such as myocardial infarction, heart failure, and arrhythmias [1]. Although air quality in the United States is relatively better than in many low- and middle-income countries, approximately 156.1 million Americans still live in areas where air pollution levels exceed federal standards. This widespread exposure contributes to an increased risk of cardiovascular complications [2,3]. Given these concerns, it is essential to understand how exposure to air pollution affects cardiac health.

Air pollution consists of a complex mixture of particles and gases, and its chemical composition is mainly influenced by its source, such as industrial emissions, vehicle exhaust, wildfires, and biomass combustion, but it is also influenced by a multitude of environmental factors/conditions, such as temperature, humidity, and other co-pollutants. Among the PM constituents, particulate matter (PM) is categorized based on aerodynamic diameter into PM10 (particles ≤ 10 µm), PM2.5 (particles ≤ 2.5 µm), and ultrafine particles (UFPs, ≤0.1 µm) [4], which are among the most studied and reactive components, and numerous studies have documented their harmful effects on cardiopulmonary, vascular, neurological, and pulmonary systems [5,6,7,8].

Fine and ultrafine particulate matter (PM) are associated with adverse cardiovascular outcomes through both direct effects on cardiac tissues and indirect systemic inflammation [9]. Among the constituents of PM, carbon black (CB), a surrogate for ultrafine particulate carbon core, has gained attention due to its widespread industrial use, high surface area for chemical adsorption, and ability to reach extrapulmonary targets via systemic circulation [10]. CB is also a possible human carcinogen and a significant environmental hazard [10]. Recently, several studies have also reported the cardiovascular, pulmonary, digestive, and cognitive impacts of chronic exposure to CB as a surrogate of ultrafine PM and burn pit particulate matter to understand its implications for human health [8,11,12,13,14,15,16,17]. However, despite its advancements in cardiovascular research, a comprehensive understanding of the molecular mechanisms driving CVD, due to this CB exposure, remains elusive.

Mitochondria, the major energy source of the cell, have been implicated in the etiology of various cardiovascular diseases [18]. The crucial roles played by mitochondria, including ATP production for cardiac contraction and relaxation, and their involvement in several metabolic processes and oxidative stress, underscore their significance in cardiovascular health. Recently, in vivo, and in vitro analyses have revealed that mitochondrial dysfunction plays a central role in the pathogenesis of pollution-induced injury, specifically under CB exposure [16,19,20]. The heart, with its high metabolic demand, is particularly vulnerable to alterations in mitochondrial function that impair energy production, redox balance, and substrate utilization [21,22]. Mitochondria dynamically regulate cardiac bioenergetics by oxidizing both fatty acids and glucose, and disruptions in this metabolic flexibility can contribute to the development of CVD [23,24,25]. However, the specific cellular mechanisms linking mitochondrial dysfunction to the promotion of cardiovascular diseases in the context of ultrafine PM exposure or CB remain inadequately understood.

Lysine acetylation, a reversible post-translational modification modulated by mitochondrial acetyltransferases such as GCN5L1 and deacetylases like SIRT3, modulates key enzymes in the tricarboxylic acid (TCA) cycle, oxidative phosphorylation (OXPHOS), fatty acid oxidation (FAO), glucose oxidation (GO), and antioxidant defense [26,27,28,29,30]. The significance of lysine acetylation is highlighted by its association with the hyperacetylation status of non-histone proteins in failing hearts [31,32,33]. Our previous work has demonstrated that increased GCN5L1 expression and hyperacetylation of mitochondrial proteins in the heart correlated with alteration of FAO and GO enzyme activity and altered ETC function under different pathological conditions, such as aging and high-fat diet (HFD) [28,29]. However, whether these mitochondrial alterations persist or worsen under CB exposure has not been systematically investigated.

In this study, we aimed to investigate the impact of repeated CB inhalation exposure on cardiac mitochondrial function, with a particular focus on fuel substrate metabolism, respiratory chain function, oxidative stress, and protein acetylation. We hypothesized that repeated CB exposure leads to hyperacetylation of mitochondrial proteins, resulting in dysregulated substrate utilization, impaired ETC complex function, increased oxidative stress, and overall bioenergetic dysfunction in the heart. Understanding these molecular mechanisms will help us identify potential therapeutic targets against this CB-induced cardiovascular toxicity.

## 2. Materials and Methods

### 2.1. Exposure System

A whole-body inhalation exposure setup was employed to deliver CB aerosols at a target concentration of 10 mg/m^3^ to mice. The exposure protocol and system components have been previously described in detail [7]. In brief, bulk CB material (Printex 90, generously provided by Evonik, Frankfurt, Germany) was aerosolized using a high-pressure acoustic generator (HPAG; IEStechno, Morgantown, WV, USA). To ensure effective deagglomeration of the ultrafine CB particles, a venturi pump (JS-60 M, Vaccon, Medway, MA, USA) was used downstream of aerosolization. Real-time monitoring of aerosol concentration was conducted with a light-scattering aerosol monitor (DataRAM, pDR-1500, Thermo Environmental Instruments Inc., Franklin, MA, USA). To characterize particle size and distribution, samples from the exposure chamber were analyzed using several instruments: an electrical low-pressure impactor (ELPI+, Dakati, Tempere, Finland), an aerosol particle sizer (APS 3321, TSI Inc., Shoreview, MN, USA), a scanning mobility particle sizer (SMPS 3938, TSI Inc.), and a Nano Micro-Orifice Uniform Deposit Impactor (MOUDI 115R, MSP Corp., Shoreview, MN, USA). The environmental parameters within the exposure chamber were maintained at animal-comfortable levels: temperature between 20 and 22 °C and relative humidity between 50 and 70%.

### 2.2. Animal Exposure

Eight-week-old male C57BL/6J mice were purchased from Jackson Laboratory (Bar Harbor, ME, USA). All animal handling and experimental procedures were approved by the West Virginia University Animal Care and Use Committee and complied with institutional ethical guidelines. Mice were housed under controlled conditions with a 12 h light/dark cycle and provided food and water ad libitum. Animals were randomly assigned to two groups: (1) air-exposed controls and (2) CB-exposed experimental group. While control animals inhaled filtered air, the CB group was subjected to aerosolized CB (10 mg/m^3^) 3 h per day for 4 consecutive days to simulate a repeated exposure scenario during an episode of pollution. Throughout the exposure period, mice were monitored for changes in body weight, food and water intake, and general health. Mice were euthanized 24 h after the final exposure for further biochemical analyses.

### 2.3. Cardiac Protein Isolation

To isolate proteins from whole-heart tissue, samples were first pulverized and then lysed in cold CHAPS buffer (1% CHAPS, 150 mM NaCl, 10 mM HEPES, pH 7.4) on ice for approximately two hours. The lysates were then centrifuged at 10,000× *g*, and the supernatants were collected for experiments, including Western blotting, co-immunoprecipitation, and enzymatic activity assays. For activity assays of Acyl-CoA dehydrogenases, tissue homogenization was performed using a modified Chappell–Perry Medium A buffer (120 mM KCl, 20 mM HEPES, 5 mM MgCl_2_, 1 mM EGTA, pH ~7.2), supplemented with 5 mg/mL fat-free bovine serum albumin. Post-homogenization, samples were centrifuged at 10,000× *g*, and the supernatants were used for enzymatic assays.

For mitochondrial-specific studies, heart tissues were processed using the QProteome Mitochondrial Isolation Kit (Qiagen, Germantown, MD, USA), following the manufacturer’s instructions. Briefly, heart tissues were minced and homogenized in the provided lysis buffer. The cytosolic fraction was removed by centrifugation at 1000× *g* for 10 min at 4 °C. The resulting pellet was resuspended in disruption buffer and centrifuged again at 1000× *g* for 10 min. The supernatant from this step was further centrifuged at 6000× *g* for 10 min to isolate mitochondria. The final mitochondrial pellet was resuspended in an appropriate buffer depending on the experiment.

### 2.4. Western Blotting

Protein expression analysis was conducted via Western blotting. Protein lysates were prepared using lithium dodecyl sulfate (LDS) sample buffer and resolved on Bolt SDS-PAGE gels (4–12% or 12% Bis-Tris; Life Technologies; Waltham, MA, USA). Proteins were electrophoretically transferred onto nitrocellulose membranes (Life Technologies) and probed with specific primary antibodies. The antibodies used included the following: rabbit LCAD (Cat #17526), rabbit Cpt1b (Cat #22170), rabbit PC (Cat #16588), rabbit SOD2 (Cat #66474), rabbit PRDX3 (Cat #10664), rabbit PRDX5 (Cat #17724), rabbit PGC1A (Cat #66369), rabbit TFAM (Cat #22586), and rabbit HADHA (Cat #10758) from Proteintech; rabbit acetyl-lysine (Cat #9814S), rabbit glutamate dehydrogenase (GDH) (Cat # D9F7P), rabbit PDP1 (Cat #D8T6L), rabbit PDH (Cat #C54G1), rabbit COX IV (Cat # 3E11), rabbit Catalase (Cat # D5N75), and rabbit Sirt3 (Cat #D22A3) from Cell Signaling Technologies; Danver, MA, USA, rabbit K122-SOD2 (Cat #214675) and rabbit PDK4 (Cat #13776), and mitochondrial OXPHOS complex proteins, including NDUFB8, SDHB, ATP5A, MTCO1, and UQCRC2, were detected using a mouse monoclonal antibody cocktail (Cat #ab110413) from Abcam; Waltham, MA, USA. The anti-GCN5L1 antibody was made by Covance and validated in prior studies [34]. Fluorescent secondary antibodies, e.g., anti-mouse or anti-rabbit IgG tagged with infrared dyes (700 nm and 800 nm; LiCor Biosciences; Lincoln, NE, USA), were used for detection. Blots were scanned using a LiCor imaging system, and band intensities were quantified using ImageJ software (version: 1.54g, NIH, Bethesda, MD, USA).

### 2.5. Immunoprecipitation

For protein acetylation analysis, immunoprecipitation (IP) was performed using heart lysates prepared in CHAPS buffer supplemented with deacetylase inhibitors: Trichostatin A (100 µM) and Nicotinamide (5 mM) to preserve acetylation status. Equal amounts of protein were incubated overnight at 4 °C with a rabbit monoclonal acetyl-lysine antibody (Cell Signaling Technology). Immune complexes were captured using Protein G-agarose beads (Cell Signaling Technology), washed three times with CHAPS buffer to remove non-specifically bound proteins, and eluted with LDS sample buffer at 95 °C. Eluted proteins were then separated via SDS-PAGE on 4–12% Bis-Tris Bolt gels and analyzed by Western blot using target-specific antibodies. A fraction of protein lysates was saved before immunoprecipitation to use as an “input” control to confirm the presence of target proteins. Densitometric quantification of bands was performed with ImageJ.

### 2.6. Blue Native Page (BN-PAGE)

To evaluate the assembly of electron transport chain (ETC) complexes I–V, blue native polyacrylamide gel electrophoresis (BN-PAGE) was performed as previously described using equal amounts of protein [28,35,36]. Briefly, 20 µg of mitochondrial protein was solubilized in 1% digitonin, followed by the addition of Coomassie Blue G-250 prior to gel loading. Samples were then loaded onto 4–16% Native PAGE gels. Wells were rinsed with dark cathode buffer (10 mL 20× Native PAGE buffer, 10 mL 20× cathode additive, 180 mL deionized water), and electrophoresis was initiated at 110–120 V. After ~30 min, dark buffer was replaced with light buffer (same composition, but 1 mL cathode additive). The run continued until adequate protein separation was achieved (~90 min). Following electrophoresis, gels were fixed in 40% methanol and 10% acetic acid for 15 min, microwaved at 1100 W for 45 s, and then destained by incubating in 8% acetic acid for 15 min. After, gels were imaged using a Native PAGE gel imaging system, and densitometry was assessed using ImageJ software (National Institutes of Health, Bethesda, MD, USA).

### 2.7. ETC Complex Activity Assays

Mitochondrial ETC complex activities (I, III, IV, and V) were measured in isolated cardiac mitochondria using spectrophotometric assays, as previously described [35,37,38]. Protein concentration was determined via Bradford assay, and samples were diluted accordingly.

Complex I activity was assessed by monitoring NADH oxidation at 340 nm in the presence of decylubiquinone. Complex III and IV activities were measured based on the reduction or oxidation of cytochrome c at 550 nm, using decylubiquinol and reduced cytochrome c as substrates, respectively. Complex V activity was measured by tracking NADH oxidation via a coupled PK/LDH enzymatic system at 340 nm. All reactions included appropriate negative controls (e.g., rotenone for Complex I, antimycin A for Complex III, KCN for Complex IV, and oligomycin for Complex V), and specific activity was calculated using extinction coefficients for NADH (6.2 mM^−1^ cm^−1^) and cytochrome c (18.5 mM^−1^ cm^−1^).

Absorbance was recorded in 96-well format, and activity was calculated using the following formula:ΔA/min = (mean working − mean control wells)Specific Activity (nmol/min/mg) = (ΔA/min × 1000)/[(ε × volume) × protein concentration].

### 2.8. Biochemical Assays

The enzymatic activity of hydroxyacyl-CoA dehydrogenase alpha subunit (HADHA) was assessed as described previously [39]. Briefly, 8 µg of heart protein lysate (prepared in 1% CHAPS buffer) was added to a 96-well spectrophotometric assay plate containing 160 µL of 50 mM imidazole buffer (pH 7.4) and 20 µL of 1.5 mM NADH. The reaction was initiated by the addition of 10 µL of 2 mM acetoacetyl-CoA. The decrease in absorbance at 340 nm, indicative of NADH consumption, was measured every minute for 15 min. HADHA activity was expressed as micromoles of NADH oxidized per minute, per 50 µg of protein, based on a standard NADH curve.

Long-chain acyl-CoA dehydrogenase (LCAD) activity was measured as described previously [28,40], with palmitoyl-CoA (60 µM each) as a substrate. Briefly, 50 µg of protein was combined with a reaction mixture containing 0.1 M potassium phosphate buffer (pH 7.4), 50 µM 2,6-dichlorophenolindophenol (DCPIP), 2 mM phenazine ethosulfate (PES), 0.2 mM N-ethylmaleimide, 0.4 mM potassium cyanide, and 0.1% Triton X-100. Reactions were conducted at 37 °C for 4 min. A total of 60 µM palmitoyl-CoA was then added, and absorbance was recorded at 600 nm over a 5 min interval. Enzymatic activity was calculated as nanomoles of substrate oxidized per minute, per milligram of protein, using standard curves.

Pyruvate dehydrogenase (PDH) activity was measured using a commercially available assay kit (DNPH-100; BioAssay Systems; Hayward, CA, USA), following the manufacturer’s recommended protocol.

### 2.9. Statistical Analysis

GraphPad Prism (v10) software was used to perform statistical analyses. A total of 5–8 animals were used for the air and repeated carbon black exposure groups. Means ± SEM were calculated for all data sets. Data were analyzed using unpaired two-tailed Student’s *t* tests to determine differences between the air and CB exposure groups. *p* < 0.05 was considered significant.

## 3. Results

### 3.1. Aerosol Characterization

Detailed characterization of CB aerosol, including detailed physical and chemical characteristics, including particle size, surface area, inherent oxidant generation potential, volatiles, and carbonyl levels, and elemental analysis, was previously published [7,8,14]. Real-time monitoring confirmed stable aerosol generation in the ultrafine size range with a SMPS/APS-based count median diameter of 82.9 nm (2.46), ELPI+ based count median diameter of 64.5 nm (2.12), and Moudi-based mass median diameter of 0.90 µ (2.60). Additionally, gravimetric assessments confirmed the desired concentration (10.33 ± 0.80 mg/m^3^). No detectable amounts of LPS were observed in the particles. The utilized Printex 90 material is 14 nm ± 4 nm primary particles of 274 m^2^/g that can agglomerate from nm to micrometer size. As shown by X-ray Photoelectron Spectroscopy, particles are devoid of any impurities and amorphous carbon particles. Finally, both volatile organic and carbonyl chemical analysis were below the assay detection limit levels of <0.02 ug/mL and 0.05 ug/mL, respectively [36].

### 3.2. Repeated CB Exposure Increases Overall Cardiac Protein Acetylation and Induces a Pro-Acetylation Phenotype

Lysine acetylation has recently been presented as a novel regulator of cardiac mitochondrial protein function in several disease pathologies, including high-fat diet and aging [28,29]. To determine whether repeated CB exposure alters lysine acetylation in cardiac tissue, we first assessed total protein acetylation. Western blot analysis of heart lysates revealed a significant increase in global lysine acetylation in the CB-exposed group compared to the air-exposed controls (Figure 1A,B). This increase correlated with a significant increase in protein expression of mitochondrial acetyltransferase GCN5L1, while the expression of mitochondrial deacetylase SIRT3 was not changed (Figure 1C–F). These findings confirm that repeated CB exposure promotes a pro-acetylation phenotype in the heart.

### 3.3. Hyperacetylation Induced by Repeated CB Exposure Leads to Decreased Fatty Acid Oxidation Enzyme Activity

The heart relies predominantly on FAO as its primary energy source, contributing to nearly 70% of ATP production under physiological conditions [22,25]. While GO serves as an alternative fuel, metabolic flexibility, which is the ability to switch between substrates based on energy demands, is crucial for maintaining cardiac function [25]. As such, to determine the effects of repeated CB exposure on cardiac fuel utilization, we measured both FAO- and GO-related protein expression and enzymatic activities in control and repeatedly CB-exposed hearts. Western blot analysis of key FAO proteins, including carnitine palmitoyltransferase 1B (CPT1b), LCAD, and HADHA, showed no significant differences in total protein levels between CB and control groups (Appendix A). Interestingly, FAO enzyme activities of HADHA and LCAD were significantly reduced in CB-exposed hearts compared to controls (*p* < 0.05), indicating a decline in mitochondrial fatty acid utilization (Figure 2A,B).

This suggests that CB exposure impairs FAO enzyme function rather than reducing its expression through post-translational modifications such as lysine acetylation, which has been previously shown to regulate its activities. As such, immunoprecipitation pulldown of FAO enzymes through acetylation-specific antibody was performed, which revealed that key FAO enzymes, LCAD and HADHA, exhibited increased acetylation levels in CB-exposed hearts compared to controls (Figure 2C–F). This suggests that repeated CB exposure impairs FAO enzyme activity, and this could potentially be regulated by acetylation, rather than overall protein expression.

### 3.4. Hyperacetylation Induced by Repeated CB Exposure Leads to Increased Glucose Oxidation, Ensuring Metabolic Inflexibility

To assess whether repeated CB exposure alters cardiac substrate utilization from FAO to GO, we examined key regulatory enzymes involved in glucose oxidation. Western blot analysis showed significantly decreased protein expression of the GO protein PDH and its positive regulator, Pyruvate Dehydrogenase Phosphatase 1 (PDP1) (Figure 3A–C). The protein expression of Pyruvate Carboxylase (PC), a precursor protein for gluconeogenesis, was not altered in CB-exposed hearts compared to controls (Figure 3D,E). However, we did not see any changes in the protein expression of Pyruvate Dehydrogenase Kinase 4 (PDK4), which is the negative coregulator of PDH (Appendix A). Although GO protein contents were decreased in the CB-exposed model, PDH activity was significantly elevated (*p* < 0.05), indicating a metabolic shift towards glucose oxidation (Figure 3L). Furthermore, immunoprecipitation pulldown with acetyl-lysine antibody demonstrated increased acetylation of PDH, PDP1, and PC (Figure 3F–K) in CB-exposed hearts compared to controls. Coupled with the observed decreases in FAO enzyme activity, these findings suggest that CB exposure disrupts normal metabolic flexibility by shifting substrate utilization from FAO to GO.

### 3.5. Mitochondrial Electron Transport Chain (ETC) Protein Content Remains Stable, but Complex Assembly and Activity Are Impaired with Repeated CB Exposure

To assess whether repeated CB exposure affects the mitochondrial ETC content, we first assessed the protein levels of representative subunits from all five ETC complexes. Western blot analysis revealed no significant changes in overall expression levels of Complex I (NDUFB8), Complex II (SDHB), Complex III (UQCRC2), Complex IV (MTCO1), or Complex V (ATP5A) in CB-exposed hearts compared to controls, indicating that repeated exposure does not alter overall ETC protein abundance (Appendix A). We then investigated whether repeated CB exposure impacts ETC complex assembly, which is critical for maintaining efficient electron flux and reducing ROS leakage. BN-PAGE analysis revealed a significant decrease in the assembly of ETC Complex III and Complex IV in CB-exposed hearts compared to the control. In contrast, Complexes I, II, and V did not show significant alterations in assembly state (Figure 4A–E).

To further assess functional implications, enzymatic activity assays were performed for individual ETC complexes. While Complex I and V activity remained unaltered, a significant decrease in activities of Complex III and IV was observed in CB-exposed mitochondria (Figure 4F–I). These reduced activities correlated with the observed decline in complex assembly, suggesting that repeated CB exposure impairs ETC function by disrupting the protein–protein interactions required for optimal ETC architecture.

We then examined whether acetylation of these complexes was altered with repeated CB exposure. Immunoprecipitation pulldown of ETC complex subunits using pan-acetyl-lysine antibody showed increased acetylation of Complexes I, III, IV, and V in CB-exposed hearts, which may potentially contribute to destabilization of supercomplex formation and activity (Figure 4J–M). Acetylation has been shown to interfere with conformational integrity between ETC complexes and their activities, supporting a post-translational regulatory mechanism behind the observed functional defects.

These findings indicate that while repeated CB exposure does not alter total ETC protein expression, it leads to acetylation-mediated destabilization of Complexes III and IV assembly, resulting in compromised mitochondrial OXPHOS capacity and potentially increased ROS production.

### 3.6. Mitochondrial Antioxidant Response Is Dysregulated Despite Unchanged Protein Expression with Repeated CB Exposure

ETC complexes are the major source of mitochondrial ROS production. We then assessed whether the redox milieu was impacted in our repeatedly CB-exposed model. To evaluate this effect on the mitochondrial antioxidant defense system, we examined the expression of key antioxidant enzymes in the heart. Western blot analysis revealed no significant changes in the protein levels of catalase, peroxiredoxin 3 (PRDX3), and peroxiredoxin 5 (PRDX5) between air- and CB-exposed groups (Appendix A). These data suggest that repeated CB exposure does not impact the transcriptional or translational regulation of these antioxidant enzymes.

Despite stable expression levels, markers of oxidative stress revealed substantial dysregulation in redox homeostasis. Specifically, malondialdehyde (MDA) levels, which are a marker of lipid peroxidation, were significantly elevated in CB-exposed hearts, indicating increased mitochondrial oxidative stress and damage to membrane lipids (Figure 5A). This paradox between unaltered antioxidant enzyme expression and increased oxidative damage prompted us to investigate the functional state of the antioxidant system.

We then focused on superoxide dismutase 2 (SOD2), a mitochondria-specific ROS scavenger responsible for the dismutation of superoxide radicals into hydrogen peroxide. Immunoprecipitation pulldown with lysine acetylation antibody did not show any robust changes in the acetylation level of SOD2 (Figure 5B). However, acetylation of SOD2 at lysine 122 (K122), a site previously shown to inhibit its enzymatic activity, was significantly increased in CB-exposed samples (Figure 5C,D). Given that K122 acetylation of SOD2 impairs its dismutase function, these data suggest that post-translational hyperacetylation of SOD2 in the K122 site compromises its activity, leading to insufficient detoxification of superoxide radicals.

Collectively, these findings demonstrate that repeated CB exposure disrupts the mitochondrial antioxidant defense system through acetylation-mediated inhibition of key antioxidant enzymes. The resulting imbalance between ROS production and scavenging may exacerbate mitochondrial oxidative damage, contributing to the broader bioenergetic dysfunction observed in the CB exposure model.

## 4. Discussion

PM_2.5_ and UFPs pose the greatest threat to human health due to their ability to penetrate deep into the alveolar space, enter systemic circulation, and reach high-perfusion organs such as the heart and brain [41]. Numerous studies have demonstrated that PM exposure can lead to oxidative stress, endothelial dysfunction, systemic inflammation, and mitochondrial injury, all of which are implicated in cardiovascular pathology [42,43].

While associations between exposure to PM and cardiovascular mortality have been reported, studies investigating the roles of individual particle exposure and the mechanisms behind their role in CVDs are relatively understudied [44,45,46]. Comparative investigations of different particulate types demonstrate that although cardiovascular toxicity is a consistent outcome, the initiating mechanisms are particle-specific. For example, diesel exhaust particles (DEP) contain redox-active organic compounds and transition metals that generate mitochondrial superoxide and hydrogen peroxide, which leads to oxidative phosphorylation impairment and endothelial dysfunction [42,47]. Silica and titanium-dioxide (TiO_2_) nanoparticles similarly cause oxidative stress but primarily through inflammasome activation, lysosomal rupture, and calcium-dependent mitochondrial permeability transition [48,49]. In contrast, carbonaceous ultrafine particles such as CB drive direct electron transfer reactions at their surfaces and activate EGFR- and Nrf2-mediated signaling pathways, resulting in both pulmonary and systemic redox imbalance [50]. Furthermore, a study by Büchner et al. showed that even non-inflammatory concentrations of ultrafine CB increase ROS, activate Src kinase, and suppress telomerase and eNOS activity in endothelial cells, leading to premature vascular senescence [51]. In addition, Hornstein et al. demonstrated that combustion-derived carbon nanoparticles delay neutrophil apoptosis via oxidant-dependent membrane rearrangements, prolonging inflammation and tissue stress [52]. Together, these studies highlight that CB exerts both metabolic and immune-regulatory toxicity, which are distinct from metal-rich or mineral dust particles.

However, Kunovac et al. in 2023 highlighted that, despite their chemical differences, various PM types ultimately impair mitochondria through excessive ROS generation and the disruption of NAD^+^/acetyl-CoA balance, creating a link between oxidative stress and abnormal protein acetylation [50]. Given that, while the effects of CB exposure have been increasingly characterized, the cardiac consequences of repeated CB exposure remain poorly understood. This is a critical gap in knowledge, especially given that real-world environmental and occupational exposures are often repetitive or sustained over days to months rather than a single event. Consistent with that framework, our present findings suggest that repeated ultrafine CB elicits an acetylation-driven mitochondrial phenotype that is distinct from the inflammation-driven responses observed for metal-rich or mineral dust particles. Furthermore, comparative studies using standardized exposure models for CB, DEP, and TiO_2_ will help determine whether acetylation-driven mitochondrial remodeling is a distinct feature of carbon-based nanoparticles or a common outcome of particulate-induced oxidative stress.

The mechanisms by which CB impacts cardiac mitochondria may involve both direct nanoparticle translocation and indirect systemic signaling. Although the fraction of CB nanoparticles that physically cross the alveolar–capillary barrier is small, their high surface reactivity and hydrophobicity enable interactions with lung lining fluids and endothelial junctional proteins such as VE-cadherin that result in increased endothelial permeability and vascular leakiness [53]. These events permit limited particle translocation into the bloodstream and initiate oxidative and inflammatory signaling cascades within the lung. Beyond direct translocation, studies also indicate that secondary oxidative and metabolic signaling from the lungs can play a role in mediating cardiovascular injury. Inhalation of oxidant pollutants such as cigarette smoke or sulfur dioxide has been shown to provoke mitochondrial oxidative stress through the Sirt3–SOD2 axis, leading to site-specific SOD2 acetylation (particularly at Lys68) and impaired detoxification of superoxide radicals [54,55]. Similar to these exposures, CB nanoparticles generate surface-driven ROS extracellularly and can activate EGFR-dependent and inflammasome-mediated pathways in bronchial and endothelial cells, releasing cytokines (IL-1β, TNF-α) that circulate systemically [53].

Taken together, in this study, we provide novel evidence that repeated exposure to CB results in increased protein acetylation, induces mitochondrial metabolic inflexibility, disrupts ETC complex stability and function, and impairs redox homeostasis in the heart mitochondria, highlighting a potential post-translational regulatory mechanism.

This exposure level is relevant for both occupational and environmental exposure to repeated CB exposure, while occupational environments sometimes exceed the limit (8 h permissible exposure limit for CB at 3.5 mg/m^3^), with reported exposures as high as 79–675 mg/m^3^ [56,57,58]. Further, 10 mg/m^3^ exposure for 3 h alone can achieve a pulmonary deposition that is comparable to a worker exposed to 3.5–4.5 mg/m^3^ over a full workday (8 h) [14], validating the translational relevance of our repeated measures. In addition, the amount of polycyclic aromatic hydrocarbon (PAH) detected in CB was very low (74.2 ng/g); it was tightly bound to the particle surface, minimizing systemic bioavailability, and it had weak DNA-damaging properties that were similar to ROS-induced DNA damage [59,60,61,62,63].

Post-translational modifications such as lysine acetylation have emerged as key regulators of mitochondrial metabolism and bioenergetics in several cardiac and other pathologies [28,29,33,64]. Mitochondria are essential organelles that maintain cellular energy homeostasis and metabolism, regulate redox balance, and modulate apoptotic signaling. Recent studies have identified mitochondria as sensitive targets of air pollution, particularly UFPs like CB, due to their ability to generate excessive ROS, impair ETC activity, increase lipid peroxidation, and disrupt mitochondrial membrane potential [7,17,42,65]. Acetylation of mitochondrial proteins can modulate enzymatic activity, affect mitochondrial metabolism and bioenergetics and alter organelle dynamics, particularly in energy-demanding tissues like the heart [26,66]. This process is balanced by mitochondrial lysine acetyltransferases, such as GCN5L1, and sirtuin family deacetylases, most prominently SIRT3 [67,68]. Notably, our findings show a significant increase in mitochondrial acetyltransferase GCN5L1 that correlated with increased lysine acetylation of cardiac proteins under repeated CB-exposure [Figure 1]. However, our findings of elevated GCN5L1 expression without changes in Sirt3 are consistent with a shift toward a hyperacetylated mitochondrial phenotype. A recent study by Dikalova et al. examining the pathogenic role of CypD acetylation in endothelial dysfunction and hypertension showed that the relative ratio of GCN5L1 to Sirt3 is a critical determinant of mitochondrial acetylation balance, where a 250% increase in GCN5L1/Sirt3 ratio promoted CypD acetylation, mitochondrial oxidative stress, and hypertension [69]. Furthermore, our study showed, for the first time, a disruption in mitochondrial metabolism with repeated CB exposure. A healthy heart is metabolically flexible in utilizing several fuel substrates for metabolism to produce ATP through the ETC complex, and over 70% of the ATP produced is from fatty acids [23,24]. In our repeated CB exposure model, we showed a significant reduction in fatty acid oxidation and a potentially compensatory shift/reliance on glucose oxidation. We show significantly decreased enzymatic activity of FAO proteins, such as HADHA and LCAD, without any changes in their protein expression or fatty acid protein import through CPT1b in our model [Figure 2 and Appendix A]. Immunoprecipitation assays confirmed that this reduction in FAO function correlates with increased acetylation of key enzymes, suggesting that repeated CB exposure induces functional inhibition through hyperacetylation [Figure 2]. This aligns with prior work showing that GCN5L1 negatively regulates FAO via direct acetylation in other cardio pathology models of high-fat diet, aging, and ischemia–reperfusion injury [28,29,34].

In terms of glucose oxidation, although the repeatedly CB-exposed hearts showed decreased protein levels of PDH and PDP1, the enzymatic activity of PDH was elevated, accompanied by increased acetylation [Figure 3]. This sustained hyperacetylation observed here in repeated CB exposure may stabilize PDH activity or reduce inhibitory phosphorylation, driving the observed shift in fuel preference from fatty acids to glucose. Such a metabolic shift is also a well-described compensatory adaptation in the failing heart to preserve ATP generation under stress [25].

The mitochondrial ETC is essential for oxidative phosphorylation and ATP production in cardiac tissue, where energy demand is among the highest of all organs. The ETC operates through a series of multimeric protein complexes (Complexes I–V) that not only transfer electrons to generate a proton gradient but also organize into supercomplexes, which enhance respiratory efficiency and reduce electron leak [70,71]. Impairment in ETC integrity has been associated with reduced ATP production, excessive ROS generation, and progression of cardiovascular pathologies, including heart failure and ischemic injury [72,73]. In our repeated CB exposure model, while total protein levels of ETC complexes remained unchanged, we observed a marked disruption in the assembly of ETC supercomplexes, particularly affecting Complex III and Complex IV. Further, we found that ETC subunits from Complexes I, III, IV, and V were hyperacetylated in CB-exposed hearts [Figure 4 and Appendix A]. Studies have shown that acetylation of Complex I subunits can reduce their NADH dehydrogenase activity [74]. Similarly, other post-translational modifications of Complex IV (cytochrome c oxidase/COX) and its electron carrier cytochrome c have been found in several common human diseases, including stroke and myocardial infarction, inflammation, including sepsis, and diabetes, where changes in COX or Cytochrome C phosphorylation/acetylation lead to mitochondrial dysfunction contributing to these diseases’ pathophysiology [75,76]. However, our study is the first to show how acetylation-mediated regulation of ETC complexes impacts repeatedly CB-exposed hearts.

Mitochondria are the principal source of intracellular reactive oxygen species (ROS), primarily through electron leakage from complexes I and III of the electron transport chain (ETC) during oxidative phosphorylation [77,78]. To counterbalance ROS, mitochondria harbor a highly specialized antioxidant defense system, including enzymes such as SOD2, catalase, and peroxiredoxins (PRDX3 and PRDX5). Among these, SOD2 plays a pivotal role in the mitochondrial matrix through the dismutation of superoxide radicals to hydrogen peroxide, which is further detoxified by catalase and PRDXs [79]. Disruption of these defenses, particularly via post-translational modification, has been increasingly recognized as a central mechanism in mitochondrial dysfunction during environmental exposures and aging [80,81]. In our repeated CB exposure model, we detected a significant increase in lipid peroxidation markers (MDA levels), indicating elevated oxidative stress. Also, we found enhanced acetylation of SOD2 at lysine 122, a well-characterized inhibitory modification that suppresses its dismutase activity [Figure 5] [82]. Hyperacetylation of SOD2 at this K122 lysine residue has been shown to impair its activity, leading to the accumulation of ROS and promoting mitochondrial damage [82]. This observation mirrors prior findings in models of cardiac ischemia–reperfusion and high-fat diet, where acetylation mediated by GCN5L1 compromises antioxidant defenses and exacerbates oxidative injury [34,83]. Overall, these findings suggest that acetylation-mediated suppression of antioxidant enzymes, rather than transcriptional suppression, is the likely driver of redox imbalance in our repeatedly CB-exposed model.

Together, our findings demonstrate that repeated CB exposure induces a sustained hyperacetylated phenotype in the cardiac mitochondria, resulting in metabolic inflexibility, ETC destabilization, and compromised antioxidant defenses. The GCN5L1–acetylation mechanism appears central to this pathophysiology and may represent a viable therapeutic target. However, further studies are warranted to strengthen the role of GCN5L1 in CB exposure. Specifically, studies looking at isolated mitochondria in a chronic exposure model will allow us to examine mitochondria-specific effects. Additionally, assessing other acetylation sites of proteins, such as the SOD2-K68 acetylation site, a lysine site reported to be a sensitive marker of oxidative stress [84] would provide further insight into acetylation-mediated redox regulation.

Mitochondrial dysfunction is central to the pathogenesis of several cardiac diseases, and dysfunctional mitochondria over time result in cardiac function deterioration, like inflammatory or fibrotic remodeling, and ultimately failure. We have previously demonstrated that acetylation-dependent mitochondrial changes result in cardiac dysfunction in aging and high-fat diet models [29,85]. As such, our next goal is to assess chronic CB exposure, integrating both echocardiography and ex vivo mitochondrial respiration assays to define how these longer acetylation-driven mitochondrial changes evolve into structural and functional cardiac pathology. Additionally, investigation into lung–heart communication via extracellular vesicles (EVs) could reveal upstream signals that propagate oxidative and metabolic damage systemically, particularly under repeated inhalation.

## 5. Conclusions

This study provides the first evidence of repeated CB inhalation-induced cardiac mitochondrial dysfunction through lysine acetylation. Repeated CB exposure impairs fatty acid oxidation, promotes a metabolic shift toward glucose utilization, destabilizes the electron transport chain, and compromises antioxidant defenses by altering protein function through lysine acetylation. These findings highlight acetylation as a critical regulatory mechanism linking environmental exposure to mitochondrial and cardiovascular pathology and make PTM-like lysine acetylation a potential therapeutic target in inhalation-related heart disease.

## Figures and Tables

**Figure 1 cells-14-01728-f001:**
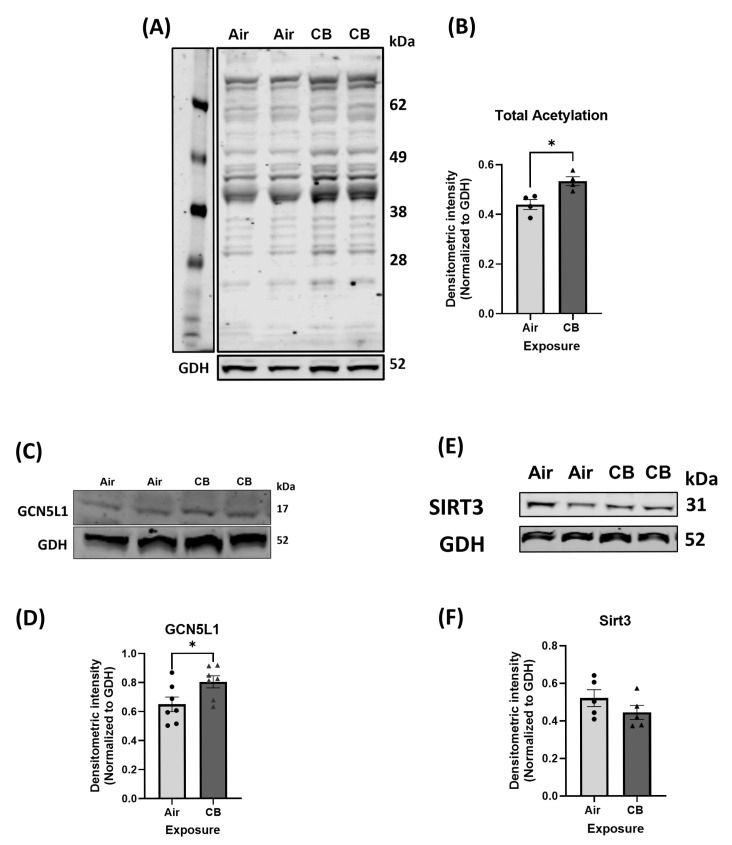
Repeated carbon black exposure increases global lysine acetylation and induces a pro-acetylation environment in the heart. (**A**) Representative Western blot showing total cardiac lysine acetylation using a pan-acetyl-lysine antibody. (**B**) Densitometric quantification of total cardiac lysine acetylation protein; *n* = 4. (**C**,**E**) Representative Western blot of mitochondrial acetyltransferase GCN5L1 and deacetylase SIRT3. (**D**,**F**) Densitometric quantification of GCN5L1; *n* = 7, SIRT3; *n* = 5 protein levels. Values are normalized to GDH and expressed as mean ± SEM; * *p* < 0.05 significance was determined using two-tailed Student *t* tests.

**Figure 2 cells-14-01728-f002:**
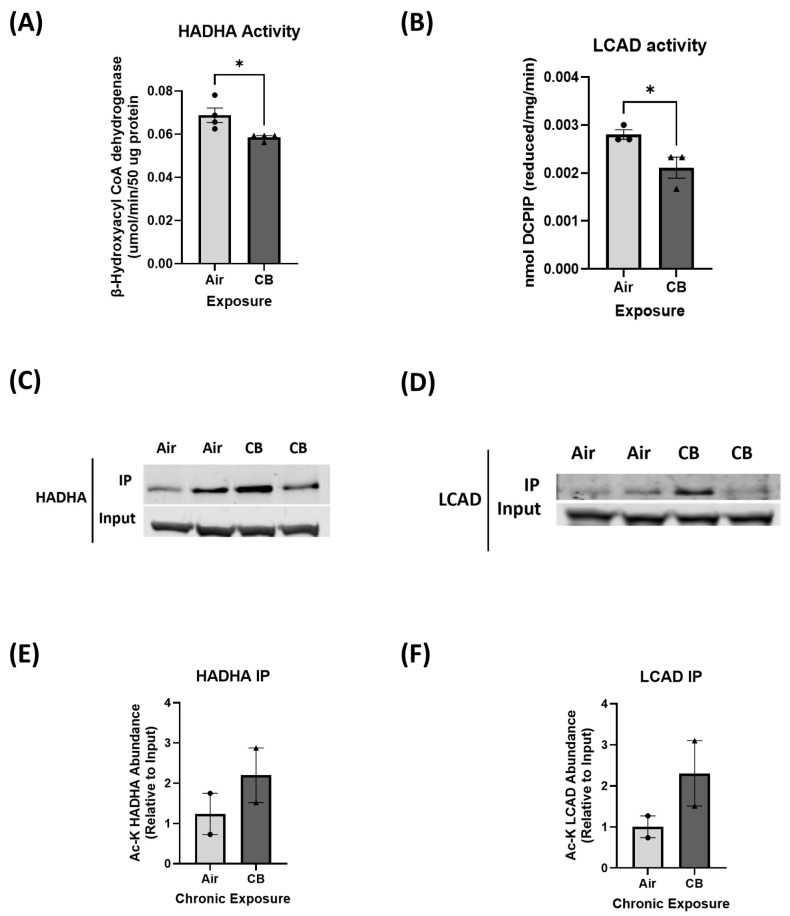
Enzymatic activity of FAO proteins is suppressed following repeated CB exposure. (**A**,**B**) In vitro activity assays for LCAD and HADHA in air- and CB-exposed hearts. (**C**,**D**) Immunoprecipitation analysis of acetylation state for HADHA and LCAD in the air- and CB-exposed samples. (**E**,**F**) Densitometric quantification of acetylated HADHA and LCAD; *n* = 2. Values represent mean ± SEM; * *p* < 0.05 vs. Air. *p*-value significance was determined using two-tailed Student *t* tests.

**Figure 3 cells-14-01728-f003:**
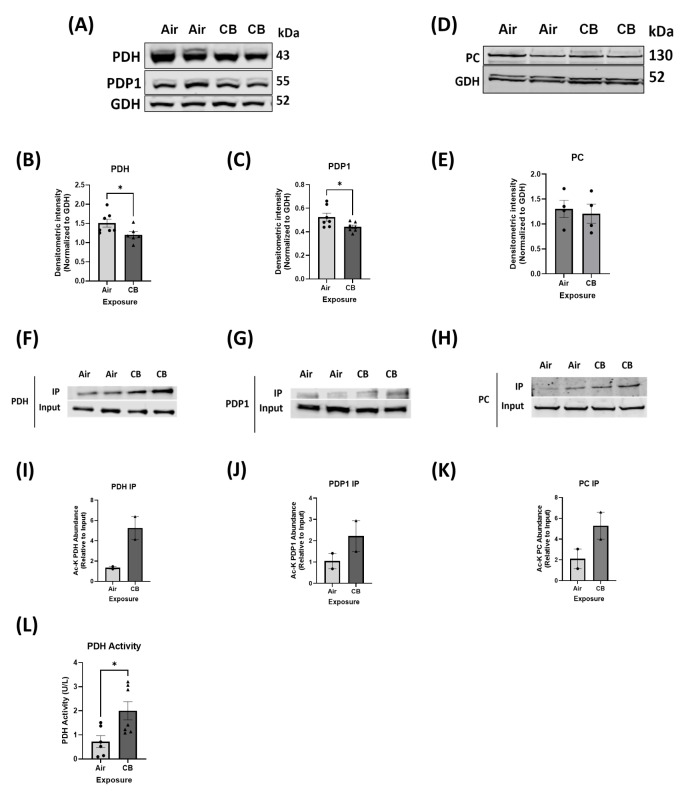
Repeated CB exposure promotes hyperacetylation and increased activity of the GO enzyme. (**A**–**E**) Western blot analysis and densitometry of PDH, PDP1, and PC protein levels. (**F**–**H**) Immunoprecipitation with pan-acetyl lysine antibody to analyze acetylation status of PDH, PDP1, and PC in the air- and CB-exposed groups. (**I**–**K**) Densitometric quantification of acetylated PDH, PDP1, and PC; *n* = 2. (**L**) In vitro activity assays for PDH in air- and CB-exposed hearts. Values are normalized to GDH and expressed as mean ± SEM; * *p* < 0.05 vs. Air. *p*-value significance was determined using two-tailed Student *t* tests.

**Figure 4 cells-14-01728-f004:**
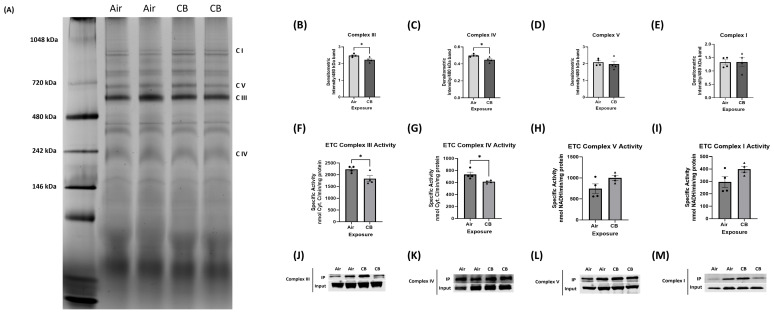
Repeated CB exposure alters mitochondrial supercomplex assembly and activity through hyperacetylation. (**A**–**E**) Blue Native PAGE analysis of Complex I, III, IV, and V supercomplexes; *n* = 4. (**F**–**I**) Enzymatic activity assays of mitochondrial ETC complexes in air- and CB-exposed hearts; *n* = 4. (**J**–**M**) Immunoprecipitation of mitochondrial ETC complexes with pan-acetylated antibody in the air and CB groups. Values are normalized to total protein input and expressed as mean ± SEM; * *p* < 0.05 vs. Air. *p*-value significance was determined using two-tailed Student *t* tests.

**Figure 5 cells-14-01728-f005:**
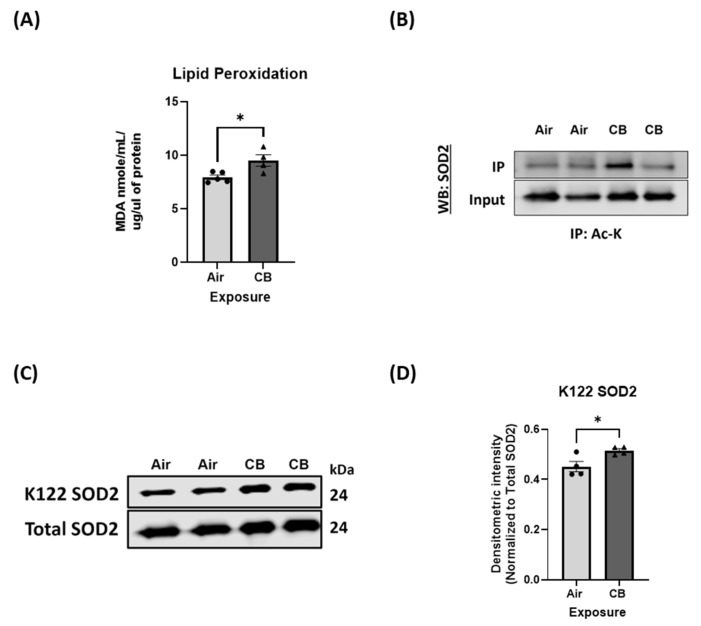
Repeated CB exposure disrupts antioxidant enzyme function through hyperacetylation. (**A**) Lipid peroxidation analysis for MDA formation in air- and CB-exposed hearts, as a marker of oxidative stress; *n* = 4. (**B**) Immunoprecipitation of mitochondrial SOD2 with pan-acetylated antibody in the air and CB groups. (**C**,**D**) Western blot and densitometric analyses of K-122 SOD2 and total SOD2 in air- and CB-exposed groups; *n* = 4. Values are expressed as mean ± SEM; * *p* < 0.05 vs. Air. *p*-value significance was determined using two-tailed Student *t* tests.

## Data Availability

All the relevant data that support the findings of this study are available within the article (and/or) its Appendix A.

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
