# Peer review of "Inhalation of Ultrafine Carbon Black-Induced Mitochondrial Dysfunction in Mouse Heart Through Changes in Acetylation"

_cells, 2025, doi:10.3390/cells14211728_

Round 1
Reviewer 1 Report
Comments and Suggestions for Authors
This is an interesting work reporting that inhalation of ultrafine carbon black induced mitochondrial dysfunction in mouse heart through changes in acetylation. Authors found very minor changes in cardiac protein expression, however, acetylation and activity of multiple mitochondrial metabolic enzymes were altered by ultrafine carbon black. They found increased lipid peroxidation and acetylation of antioxidant enzyme SOD2 at the K-122 site reducing enzymatic activity and contributing to oxidative stress. Authors suggest potential role of key post-translational mechanism in environmental particulate exposure to mitochondrial impairment and provide a potential therapeutic target for carbon black induced cardiotoxicity.
The following minor changes are recommended:
1) mice were exposed to carbon black for 4-days, however, it is not clear how many days after exposure mice were euthanized. Add this information to method section.
2) Add to Figure 1 (A) the molecular weights markers or indicate the molecular weights of key bands.
3) Please discuss the imbalance between GCN5L1 and Sirt3 (ratio) as a driver of mitochondrial acetylation as recently described (PMID: 38639088).
4) Add quantifications to Figure 2 (C) and Figure 2 (D).
5) Add Quantification to Figure 3 (F). Explain why increased PHD acetylation (reducing PDH activity) apparently associated with increased PDH activity in Figure 3 (I). Check this assay and and explain WHY PDH activity may be increased.
6) Correct "(F)" to "(I)" in Figure 3 legend.
7) Add index (A) next to image in Figure 4.
8) Add discussion regarding the potential limitations of the experimental approach. This may include the lack of the studies on isolated mitochondria which can increase the protein abundance and provide more robust results (a), Lack of SOD2-K68 acetylation analysis which nay better reflect oxidative stress (PMID: 28684630).
9) Please add to Discussion section potential mechanism "How the Inhalation of ultrafine carbon black can reach the heart?" See potential suggestions in PMID: 30608177 and PMID: 38410870.
Reviewer 2 Report
Comments and Suggestions for Authors
The authors proposed the manuscript entitled “Inhalation of ultrafine carbon black induced mitochondrial dysfunction in mouse heart through changes in acetylation” as research article for publication in the journal Cells. Using a mouse model, they investigated the effects of repeated inhalation on the cardiac mitochondrial function, exploring the metabolic pathways and regulatory mechanisms involved in energy production. Assessing different biochemical assays, the authors observed significant changes in fatty acid oxidation, glucose oxidation, and disrupted electron transport chain, other than hyperacetylation of mitochondrial proteins.
Although the research provides a potential role of post-translational mechanisms in environmental particulate exposure to mitochondrial impairment, providing therapeutical target for cardiotoxicity induced by air pollution, there are several points which could be improved.
Following, there are my comments:
- The principal limitation of the study is its focus on only the 24-hour post-exposure timepoint. Exploring only the 24-hour timepoint is problematic since cardiovascular effects from pollution typically develop over months to years, potentially missing the most clinically relevant pathological processes.
- The absence of positive controls (such as known cardiotoxic substances) probably makes it difficult to understand the real effect size and clinical relevance of the findings. Furthermore, without comparing carbon black with other ultrafine particles, it is not possible to determine if the reported effects are specific to carbon black or indicative of general nanoparticle toxicity.
- The authors need to be clearer about the type of t-test performed for the different comparisons. It must be explained if they used multiple comparison corrections, since this aspect is not specified in the text.
- The study delineates molecular alterations without sufficiently correlating them to physiological relevance. The gap between biochemical measurements and functional cardiac results reduces the mechanistic understanding and therapeutic significance of the findings.
- The discussion misses considerations concerning other bodies of research in the field with conflicting results or exploring the effects of other particles. Since pollution exposure in the real world is complicated and involves several types of particles, it is important to consider the carbon black-specific findings in relation to other studies analysing the toxicity effects of particulate matter.
Round 2
Reviewer 2 Report
Comments and Suggestions for Authors
The authors clearly addressed all the points raised by the reviewers, responding to critical points and modifying the parts of the manuscript that needed improvement.
The current version of the manuscript is much more readable and clear to the reader.